# Multiscale Simulations for Defect-Controlled Processing of Group IV Materials

**Gaetano Calogero** [1,*] , **Ioannis Deretzis** [1] , **Giuseppe Fisicaro** [1] , **Manuel Kollmuß** [2] , **Francesco La Via** [1] , **Salvatore F. Lombardo** [1] , **Michael Schöler** [2] , **Peter J. Wellmann** [2] and **Antonino La Magna** [1,*]

1   Consiglio Nazionale delle Ricerche, Istituto per la Microelettronica e Microsistemi CNR-IMM, Zona Industriale VIII Strada 5, 95121 Catania, Italy
2   Crystal Growth Lab, Materials Department 6 (I-Meet) Friedrich-Alexander-Universität Erlangen-Nürnberg, Martenstraße 7, 91058 Erlangen, Germany
*   Correspondence: gaetano.calogero@imm.cnr.it (G.C.); antonino.lamagna@imm.cnr.it (A.L.M.)

**Abstract:** Multiscale approaches for the simulation of materials processing are becoming essential to the industrialization of future nanotechnologies, as they allow for a reduction in production costs and an enhancement of devices and applications. Their integration as modules of "digital twins", i.e., a combined sequence of predictive chemical–physical simulations and trained black-box techniques, should ideally complement the real sequence of processes throughout all development and production stages, starting from the growth of materials, their functional manipulation and finally their integration in nano-devices. To achieve this framework, computational implementations at different space and time scales are necessary, ranging from the atomistic to the macro-scale. In this paper, we propose a general paradigm for the industrially driven computational modeling of materials by deploying a multiscale methodology based on physical–chemical simulations bridging macro, meso and atomic scale. We demonstrate its general applicability by studying two completely different processing examples, i.e., the growth of group IV crystals through physical vapor deposition and their thermal treatment through pulsed laser annealing. We indicate the suitable formalisms, as well as the advantages and critical issues associated with each scale, and show how numerical methods for the solution of the models could be coupled to achieve a complete and effective virtualization of the process. By connecting the process parameters to atomic scale modifications such as lattice defects or faceting, we highlight how a digital twin module can gain intrinsic predictivity far from the pre-assessed training conditions of black-box "Virtual Metrology" techniques.

**Keywords:** digital twin modules; process simulation; multiscale modeling; Kinetic Monte Carlo; finite element methods

## 1. Introduction

The digitalization of materials processing through numerical simulations is a consolidated paradigm of advanced future nanotechnologies. The ideal framework implies the presence of "digital twin modules" for any processing machine in the manufacturing lines of an industrial environment [1–3]. Such modules should virtually reproduce the action of the equipment and should be integrated in decisional frameworks, including experimental datasets [4,5]. The benefit of virtualization is clear both for the optimal application of the processing, as well as for the remote control and use of the equipment. The integration of the process digital twin modules into the full production line is generally managed by higher level Artificial Intelligence (AI) to realize a fully fledged digital twin of the factory ecosystem and related market [1,2].

In a fully digitalized process line, each process should be supported by a digital twin module. The module could belong to two generic categories: (1) black-box methods also known by the term "Virtual Metrology" (VM), (2) physical–chemical simulation methods.

The first category (see Refs. [6–8] for some examples of specific applications in semiconductor manufacturing) uses historical datasets, including: process parameters, sensor data, real metrology measurements, etc. (Figure 1), to infer by means of statistical methods (and, recently, machine learning techniques) the process outcomes in terms of modification of the incoming system [6–10]. VM has no intrinsic predictivity, and the success of its predictions relies on the quality of the training dataset and on the alignment between the current process conditions with the ones generating the previous process data. Physical–chemical simulations, the main topic of this paper, have a similar scope, but are based on the "in-silico" reproduction of the non-equilibrium kinetics induced by the processes. Therefore, after appropriate calibration, they can predict the process results far from the pre-assessed conditions.

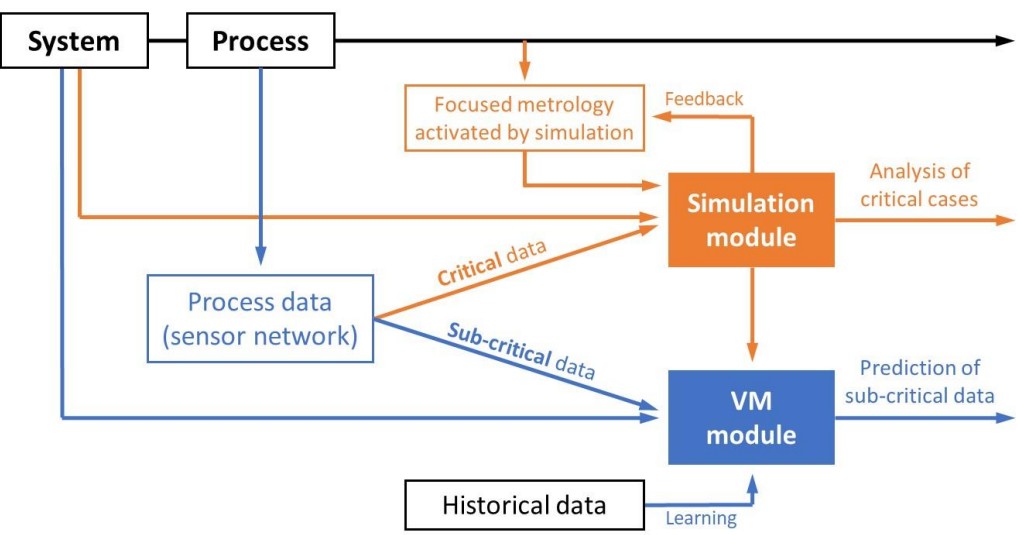

**Figure 1.** Schematic of a digital twin module integrating both physical–chemical simulations and virtual metrology (VM) tools for a given process. The "sub-critical data" refer to the standard operation of the process machine in the manufacturing chain, the "critical data" refer to anomalous deviations of the output from the process expectations or the output in a development stage (due to, e.g., a new required specification for the process). The focused metrology can be also destructive, while process data are obtained with noninvasive methods.

According to the availability of computational codes and the internal choice during the manufacturing line setup, one of the two categories can optionally be considered. However, a digital twin module (see Figure 1) integrating both physical–chemical simulations and VM tools for a given process could represent the optimal solution for both the advanced process control and the development stage of a new product, e.g., by means of Design of Experiments (DoE) methodologies. On the one hand, in a process control flow, where a set of variables is continuously monitored to check the process' stability, VM allows the reduction of the real metrology steps after the process, thus improving the manufacturing yield. The reduced number of metrology results, defined as "sub-critical data" in Figure 1, can be compared with the VM predictions to validate the control procedures and, after successful comparison, integrated with the historical dataset. In case of failure of this control step (e.g., measured variables falling outside a well-established confidence interval) the data are regarded as "critical" according to the scheme in Figure 1. In this case, the manufacturing flow is stopped, and simulation methods could be activated to understand the issues. On the other hand, in a development stage in which, due to new design requirements, the machine is set to different conditions with respect to its historical use, the use of VM is of course unfeasible, and the simulation module finds its specific applications (i.e., in this case the data are always considered "critical").

This ideal scenario is far from being effectively achieved, and some current simulation-type versions of digital twin modules barely or partially reproduce the machine functionality, especially for processes that intrinsically involve complex kinetics at many interconnected scales (macro, meso, nano, atomic) [11–13]. In particular, if a process aims at the manipulation of a system with nanoscale resolution, the module should predict, with atomistic detail, the transformation of the system from the initial to the final state with respect to the controllable process parameters (i.e., the equipment setting) [14]. This nanoscale transformation should be able to interact, in a seamless multiscale fashion, with meso or macroscale models in order to predict the impact of such atomistic effects on larger length scales and over the real time scale of the process. Only when these requirements are fulfilled can the eventual multiscale simulation module be integrated in the digital twin of the production line [5,11].

In this paper, using two completely different examples, we will discuss the computational development that is necessary for the achievement of a reliable digital twin module of two nanoelectronic processes. We believe that the method strategy is generalizable to a broad set of cases, and we will try to indicate how the approach can be applied under other development conditions.

In particular, we will focus on the twinning procedures in the case of Pulsed Laser Annealing (PLA) and Physical Vapor Deposition (PVD) in a Physical Vapor Transport (PVT) setup. We notice that with respect to the materials being processed, the equipment can be considered as an external environment which drives the microscopic system in non-equilibrium thermodynamic conditions for the time frame of the process duration. In the case of pulsed laser annealing, non-equilibrium is the ultra-fast heating/quenching caused by the absorbed pulsing electromagnetic energy; while in the case of PVD processes, it is the supersaturation of the vapor phase components close to the material surface with respect to the solid/vapor equilibrium densities.

With no significant detriment of generality, atomic structures of group IV materials will be considered when the discussion reaches the atomic scale of the multiscale methods. Of course, this choice is justified by the extreme technological interest in group IV materials (C, Si, Ge, SiGe, SiC). Anyhow, we believe that the arguments discussed here are also of broad interest for materials scientists who do not focus their research on group IV-based systems.

This paper is organized as follows: First, the methods and simulation tools are introduced in Section 2. Environmental scale, mesoscale and atomic scale modeling methods are discussed in Sections 3.1–3.3, respectively, each split into two sub-sections dedicated to the two case studies. Section 4 is dedicated to the conclusive remarks.

## 2. Methods

Several methods and tools have been exploited to perform the simulations described in this paper. We introduce them briefly below, along with all relevant references. Additional theoretical and computational details will be given in Section 3 when needed.

The tool COMSOL Multiphysics [15] based on the finite element method (FEM) was used for the thermal field analysis of SiC PVD processes, adopting the material database as in Ref. [16] by Steiner et al. for all simulations.

The open-source MulSKIPS Kinetic Monte Carlo (KMC) code is used to perform PVD and PLA simulations with atomistic resolution, with the aid of the Python interface "pymulskips" distributed with it [17,18]. The code implements a calibrated superlattice KMC formalism which models PVD, PLA and also chemical vapor deposition processes for elements, alloys and compounds characterized by $sp^3$ bond symmetry, with the possibility of including both cubic or hexagonal phases, as well as point or extended lattice defects in the simulations. It also implements an I/O interface to Technology Computer-Aided Design (TCAD) tools. Global models for local growth rate predictions for SiC PVT growth are implemented and analyzed with Jupyter Notebooks available at the MulSKIPS repository [17], which are also used to set up and analyze the KMC simulations.

The Dolfin Python interface of the open-source FENICS computing FEM platform [19,20] is used to implement, with Application Programming Interfaces (APIs) written in the same programming language, the macroscale coupled method that self-consistently solves the Maxwell equations, the Fourier equation and the phase-field in a 1D, 2D or 3D mesh, which is needed for the continuum PLA process simulations. Such a platform is also used for the phase-field modeling of mesoscale silicide transformation.

A multiscale interface implemented in MulSKIPS is used to perform PLA simulations with local atomistic resolution, by replacing phase-field with superlattice KMC to model solid–liquid phase transitions. The computational details of such an integration procedure are fully described in Ref. [21].

The software Paraview [22] is used for mesh visualization and post-processing of the FEM simulations, while V_Sim is used to visualize the atomic structures [23]. All the input TCAD geometries and their mesh are constructed using the gmsh software [24,25].

## 3. Results

### 3.1. Environment Scale Study and Modeling

In most cases, the environment acting on the micro-structure under process has the characteristics of a macroscopic system, and the duration of the process can be easily extended to seconds/minutes/hours. In this case, the twinning procedures need a computational approach adequate to the macroscale of the system. Apart from some relevant exceptions in which particle tracking can be necessary for the simulation of some portions of the equipment (for example, the pre-sheath and sheath region of a plasma-aided process) [26–28], continuum models, based on (ordinary or partial) differential equations, of the evolving or stationary (but still machine-setting dependent) average fields in the machines' volumes are appropriate. This is especially true for the two case studies discussed in this subsection, where the environmental variables under consideration, e.g., temperature or electromagnetic field, vary over length scales that are inaccessible to atomistic models (larger than $10^{-3}$ m).

Sequential or parallel (self-consistent) coupling of the environmental model to the kinetic model simulating the microstructural kinetics can be applied according to the problem under study. The PVD and PLA examples are opposite situations: sequential coupling is, in general, sufficient for the PVD case, while PLA requires strict self-consistent coupling between the environmental and material variables [29].

Another issue which merits attention is the difficult validation of the digital twin module as a whole: intermediate comparisons between model and real systems should be considered at the different scales. For this scope, due to the impossible experimental characterization of the continuum variable in the 3D volume of the machines, measurements' probes have to be integrated in the equipment, allowing the dynamic point or average measurements of the fields or field-related physical quantities. Such measurements are necessary for the process control and the validation of the simulation results, which in general give more complete information on the field evolution and distribution in the volume of the equipment. In the next sub-sections, these topics are specifically treated for the PVD and PLA cases.

### 3.1.1. Temperature Field Analysis in PVD Processes of SiC Growth

In a PVD growth process, high-quality thin films are grown using atomic or molecular directional flows of the same species constituting the corresponding vapor phase of the material film. These vapor flows, in equilibrium with the solid powder at the source higher temperature, are controlled by means of the temperature and/or pressure gradients from the source to the seed region, which are implemented in the chamber. Knudsen cells are often exploited as sources, and ballistic to diffusive flow of the components can characterize the growth processes when varying the chamber pressure [30,31].

Here, we consider a PVT setup referred to as close-space PVT (CS-PVT), which is used to grow high-quality cubic silicon carbide. A detailed description of the used setup,

as well as the growth process itself, can be found in Ref. [32]. In order to predict the kinetic behavior, the thermodynamic conditions and temperature distribution inside the hot zone need to be well known. However, due to the high temperatures ($\geq$2000 °C) used in the growth setup, the acquisition of the necessary data is complicated. Numerical simulations are frequently used to solve this issue. Nevertheless, the right choice of the materials database at elevated temperatures can be challenging. The biggest problem is the uncertainty or even lack of reliable data of the temperature dependence of the heat and electrical conductivity, especially for the used graphite parts and isolations. For all simulations, the tool COMSOL Multiphysics was used (see Section 2). A 2D axisymmetric geometry was chosen, which reflects the reality of the rotationally symmetric PVT setup sufficiently well. Conduction and radiation were considered for the heat transport, whereas convection was neglected due to the small influence on the temperature field [33]. The inductive heating was calculated using the Maxwell equations. All simulations were time independent, and the transport of the material was not implemented for the determination of the temperature field. Simulations were conducted for two setups with 50 mm and 100 mm substrate diameter, respectively, and compared to temperature measurements during growth experiments. The temperature measurements were carried out with a Metis M311 quotient-pyrometer from *Sensortherm* (error: 0.3%) on top of the crucible.

Figure 2 shows the simulation results for the hot zone of the 100 mm setup, as well as an enlarged view of the sublimation sandwich zone with the axial and horizontal temperature distribution. The sandwich itself consists of a tantalum foil (the thinnest layer in the bottom panel of Figure 2), a SiC solid source above it, a graphite spacer (on the right side in the bottom panel of Figure 2) and a cubic SiC seed on top, where crystal growth takes place. The space between the seed and the source is kept small (1 mm). The resulting axial temperature gradient over the whole growth cell is high ($\Delta T_{axial} = 50 \ldots 70$ K cm$^{-1}$), favoring the growth of the cubic polytype. At the same time, the radial gradient is kept low with a maximum value of $\Delta T_{radial,seed} = 0.8$ K cm$^{-1}$. The main driving force behind the growth of SiC is the temperature gradient between the source and the slightly cooler seed. This axial gradient increases from approx. 5 °C in the center of the cell to approx. 7 °C near the edge of the seed, respectively. Once the spacer is reached, the gradient drops due to the high thermal conductivity of the graphite spacer. This temperature distribution arises from the design of the growth cell. The system is heated inductively using a copper coil positioned symmetrically around the crucible. Additionally, the cooling channel is placed directly over the center of the crucible. Therefore, the hottest point of the crucible is not located in the center but is shifted towards the crucible edges, leading to the described temperature profile.

The temperatures from the simulation were used to calculate the vapor pressures of the gas species inside the growth cell by the approach of Avrov [34]. Based on the assumption of a silicon-rich gas composition, SiC$_2$ will act as the growth-limiting gas species. Therefore, the calculations were made only for this species using the equation mentioned in Ref. [35]. The supersaturation follows the same trend as the axial temperature gradient. It increases from the middle towards the edge of the seed before dropping once the spacer is reached. This behavior corresponds to the fact that a higher temperature gradient leads to higher supersaturation.

Figure 3 compares the modeled temperatures for the 50 mm setup at the crucible top and the measured temperatures at this position during growth experiments. The graph shows a nice agreement between the theoretical and experimental data. For the 100 mm setup, a similar trend could be observed (not shown). To achieve this agreement, a little adaption had to be made to the base material dataset mentioned previously. In addition to a correction factor for energy losses of the inductive heating system, the altering of the used graphite isolations has to be considered. As the parts are exposed to multiple growth runs, their thermal isolation behavior changes, leading to higher thermal conductivity and increased thermal losses. To compensate for this effect in the simulations, a space-dependent pre-factor was introduced locally for the temperature-dependent calculation of

the thermal conductivity of the insulating regions in the simulated model. However, the decrease in the thermal isolation effect cannot be applied homogeneously over the whole setup. Most inner isolations are in direct contact with the hot crucible. They will not only alter faster but will also have a greater increase in thermal conductivity compared to the outer isolations. This outer part will keep their isolation performance for a longer time as they see lower temperatures during the growth runs. This feature has to be kept in mind leading to a necessary monitoring of the operating time of the isolations and different values for the pre-factors. Using this approach, we were able to reduce the difference between simulation and reality to a value below the measurement accuracy of the used pyrometer. A possibly better agreement between simulated results and experimental measurements could potentially be achieved by means of more accurate measurement settings and further optimization of the aforementioned compensating pre-factors.

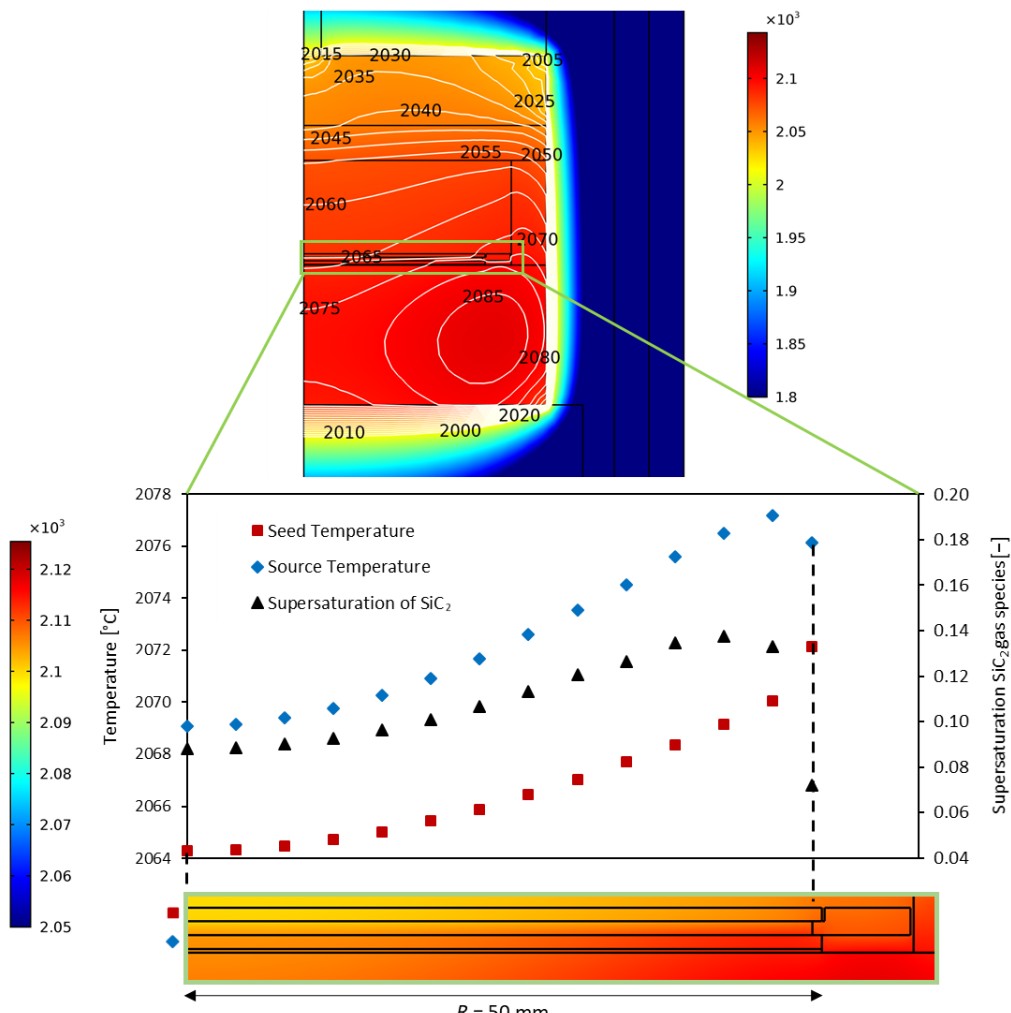

**Figure 2.** Simulation of the temperature field for an input power of 4.2 kW and calculation of the supersaturation in a vapor phase growth reactor for 100 mm 3C-SiC single crystals (the radius R is indicated in the bottom panel). The calculations of the supersaturation were based on the work of Avrov [34].

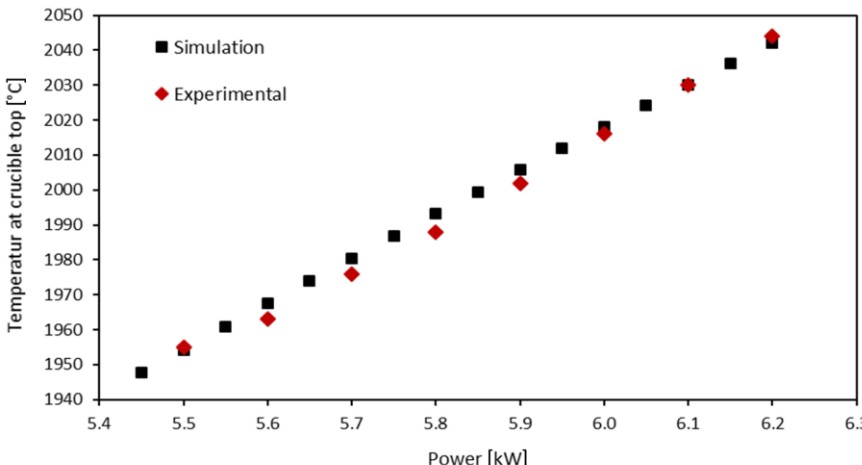

**Figure 3.** Comparison between measured temperatures at the crucible top during growth experiments in the 50 mm CS-PVT setup and simulated temperatures using COMSOL Multiphysics.

3.1.2. Ultra-Fast Heating of Materials in PLA Processing

In PLA, the environmental variables are the evolving coupled electromagnetic and thermal fields to be simulated in a suitable large volume, including the irradiated systems; and, of course, in this sense, some similarities emerge naturally with the PVD case, but the time scales are completely different. The proper description of heating dynamics in laser irradiation represents one of the key issues for the reliable simulation of actual processes. The simulation setting depends on the different process settings (continuum wave or pulsed mode, pulse duration, fixed or scanning mode; for a complete view, see Ref. [36]). The particular laser process discussed here is characterized by laser pulses with pulse durations from tens to hundreds of ns, fixed frequency and fixed beam with imping areas of the order of $\approx$ cm$^2$).

For pulse $\Delta t_{pulse}$ duration longer than 1 ns and photon energy of the order of 1 eV, $\Delta t_{pulse}$ is usually much larger than the inverse of the frequency

$$\Delta t_{pulse} \gg \nu^{-1} = (c/\lambda)^{-1} \tag{1}$$

where $\lambda$ is the laser wavelength and $c$ is the speed of light; in this condition, we can correctly use the time-harmonic solution of the Maxwell equations in order to evaluate the resistive heat averaging the "ultra-fast" time scales of the oscillating electromagnetic field [37]. Maxwell equations are coupled self-consistently with the time-dependent solution for the temperature field

$$C[\boldsymbol{r}, T(\boldsymbol{r};t)]\frac{\partial T(\boldsymbol{r},\,t)}{\partial t} = \nabla \cdot \{k_{bulk}[\boldsymbol{r}, T(\boldsymbol{r};t)]\nabla T\} + S(\boldsymbol{r},t) \tag{2}$$

by means of the laser heat source $S(\boldsymbol{r},t)$ field:

$$S(\boldsymbol{r},\,t) = \frac{\epsilon_2[\boldsymbol{r}, T(\boldsymbol{r},\,t)]}{2\varrho[\boldsymbol{r}, T(\boldsymbol{r},t)]}|\boldsymbol{E}_{t-h}(\boldsymbol{r},t)|^2 \times P_{norm}(t) \tag{3}$$

Here, $C$ is the thermal capacitance, $k_{bulk}$ is the bulk thermal conductivity, $\epsilon_2$ is the imaginary part of the complex dielectric function $\widetilde{\epsilon}$ of the heated material, $\varrho$ is the local density of the materials, $\boldsymbol{E}_{t-h}$ is the time harmonic electric field and $|\boldsymbol{E}_{t-h}|^2$ represents the intensity of the electromagnetic field. $P_{norm}(t)$ is the process-dependent normalized power density of the process laser which tends to zero within the $\Delta t_{pulse}$ time scale. We notice that with the symbol $[\boldsymbol{r}, T(\boldsymbol{r};t)]$, we indicate that any material property depends explicitly on the position and implicitly on time, through the dependence on the local temperature. Of course, due to the dependence of $\widetilde{\epsilon}$ on $T(\boldsymbol{r};t)$ (and the eventual phase transitions),

the self-consistent time harmonic solutions of the electromagnetic field vary adiabatically with $t$. For diffusive thermal transport, Equation (2) is a good approximation for many applications, but model modifications are necessary for including ballistic corrections in the temperature solutions [38–40].

The solutions of environmental variables in PLA have, in general, non-trivial behavior which corresponds to a non-linearity (and non-monotonicity) of the observed response of the real systems to the process parameters. These features make the use of a digital twin almost strictly necessary for the optimization of this process.

In Figure 4, an illuminating example of a non-linear behavior is shown (see Section 2), plotting the maximum temperature obtained as a function of the pitch and the polarization of the laser light, in simulated irradiation processes of silicon "Fin" shaped Field Effect Transistor (FinFET) array structures with fixed pulse shape ($\Delta t_{pulse} = 160$ ns) and energy density (0.6 J/cm$^2$). Such arrays with varying pitch are modeled using periodic cells with varying size along the direction transverse to the fin. In particular, we can observe that the irradiation with Transverse Electric (red line) polarization (with respect to the plane of the FinFET sections) shows a large variation in the response with the maximum temperature, which also surpasses the melting point for crystalline Si in intermediate values of the pitch for the range considered. A qualitatively similar non-monotonic trend is observed for the mixed polarization case (green line) but with the maximum temperature remaining below the melting point, while it decreases monotonically with the pitch size in the Transverse Magnetic case (blue line).

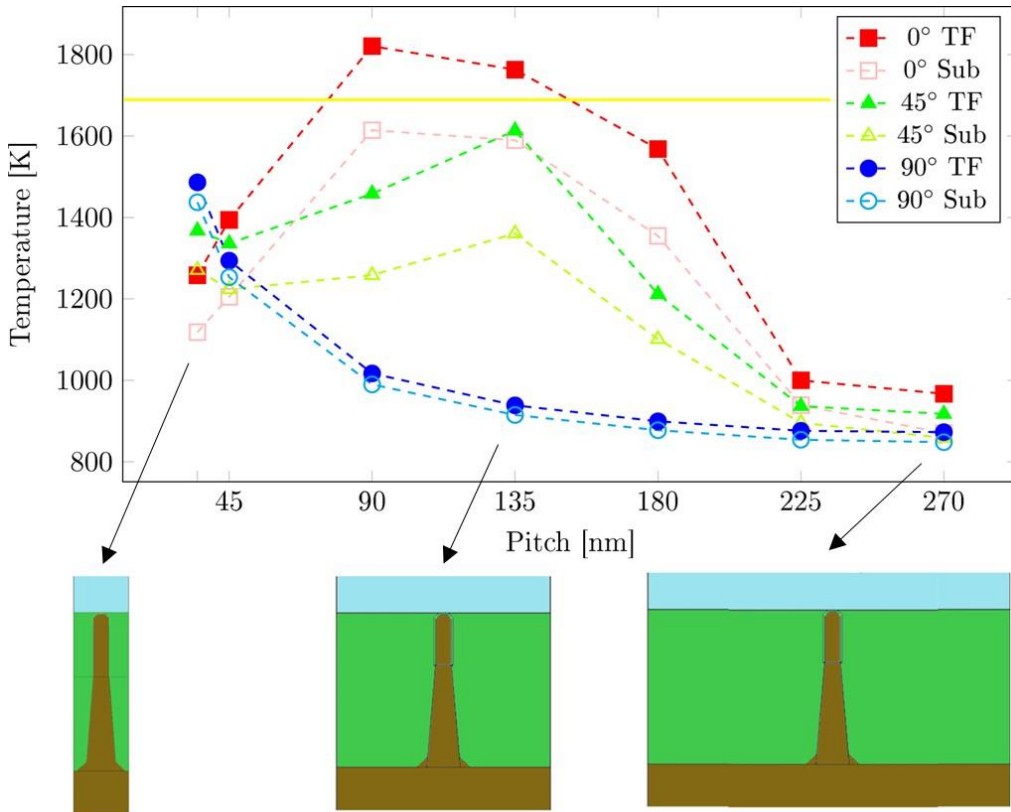

**Figure 4.** Maximum temperature (top panel) obtained in a thermal simulation of a $\Delta t_{pulse} = 160$ ns PLA with 0.6 J/cm$^2$ energy density ($\lambda = 308$ nm) in the top fin region (filled markers) and the substrate region (empty markers) of a FinFET array structure as a function of the pitch and the polarization of the laser light from Transverse Electric (TE = 0°) to Transverse Magnetic (TM = 90°). The bottom panel shows some systems with representative pitches (35 nm, 135 nm, 270 nm). Light blue, green, and brown indicates Air, SiO$_2$ and Si regions, respectively. The yellow line in the top panel indicates the melting threshold for Si.

The example presented here clearly demonstrates the ineffectiveness of a purely experimental approach to the PLA process design, which would not even be able to predict the correct trend of an environment-scale quantity such as temperature. Moreover, the experimental monitoring of the evolving field is also more difficult, due to the pulse time scale, than for the PVD/PVT case. In this case, an acceptable option for the process control which can be used for the intermediate validation of the digital twinning procedure is the in situ time-dependent measurement of transient reflectivity (TR) of the samples using an additional low-power laser as a probe (in general, at a different wavelength of the heating laser) [41]. The reflectivity may depend on the temperature and material phase fields (therefore, it is effectively a combined average probe of several evolving fields) and it can be directly compared with the analogous variables estimated by simulations. In Figure 5, an example of comparison between real and virtual in situ reflectivity is shown for an PLA process of a sample exposing an amorphous ($\alpha$) Ge surface. In the process condition, a very rapid ("explosive" see ref. [42]) phase transformation occurs in $\alpha$ Ge close to the surface during the irradiation from amorphous to liquid and then to (poly)crystalline phases, which can be monitored by TR. As can be noticed, virtual and real measurements agree [42]. We note that, while the simulated reflectivity values vary from 0 to 1, the TR setting does not allow the absolute measurement of reflectivity; therefore, only relative comparisons are relevant. In this particular case, the onset temperature of this behavior is the melting point (965 K) of $\alpha$ Ge, which is reached by the sample after about 180 ns from the starting time of the measurement.

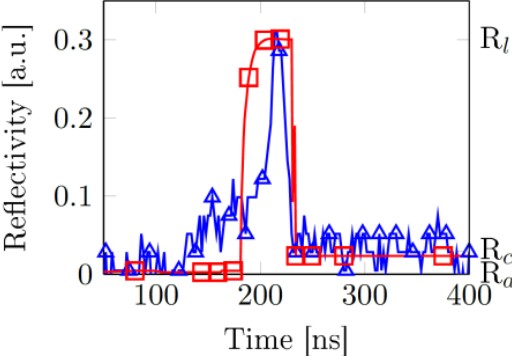

**Figure 5.** Simulated (red lines and squares) and measured (blue line and triangles) in situ transient reflectivity measurements for a $\Delta t_{pulse}$ = 160 ns PLA process with 0.56 J/cm$^2$ energy density ($\lambda$ = 308 nm). The initial system is a $\approx$ 50 nm thick amorphous Ge film on a large crystalline Ge substrate. $R_l$, $R_c$ and $R_a$ indicate the relative reflectivity level for the liquid, crystalline and amorphous phases of the material, respectively. The additional laser probe for the reflectivity measurements has a 635 nm wavelength [42].

### 3.2. Process Simulation Predictions at the Mesoscale

As discussed, the digital twin of the processing should necessarily deal with the virtual reproduction of the system kinetics. For many applications, coarse process control is sufficient, and modeling of average quantities satisfies such less stringent requirements. Mesoscale material modeling allows the simulation of relatively large systems (from the micron scale up to the sample scale) solving models for the time-dependent characterizing quantities at the feature scale. Continuum-type or particle-type alternative approaches can be applied, but they usually share the equivalent description of underlying physical/chemical phenomenology. For example, in-cell (discrete/particle) stochastic simulation or level-set (continuum) simulation can be applied with similar accuracy to model surface profiles in etching process at the mesoscale [43–45]. For the two cases studied in this paper, the mesoscale simulation is obtained with continuum methods (see Methods). Of course, for the consideration already reported, equivalent discrete modeling can be obtained.

### 3.2.1. Local Growth Rate Prediction for a 3C-SiC PVT Growth

A global model for the evaluation of the process results in terms of material growth rate can be obtained from the estimates of the mass transfer rate from seed to the substrate once the temperature field is evaluated via the chamber simulation discussed in the previous section [46]. Assuming that ballistic transport conditions occur for the Si-C molecules [47], which sublimate at a different rate at the two interfaces, approximate estimates of the growth rate for the 3C-SiC in the different positions of the growing substrate (in a fully symmetric configuration) can be obtained from the balance between the atomic species effective deposition fluxes ($j_{dep}$), coming from the sources and ruled by the source temperature $T_{Source}$, and the evaporation flux ($j_{ev}$), ruled by substrate local temperature $T_{Sub}$. Due to the particular composition of the SiC vapor pressure, Si-rich conditions are usually assumed (see Avrov et al. [34]): the gas Si species is "always" available close to the surface, and it reacts with C-containing gas species to release Si-C at the solid boundary. In these conditions, the growth rate can be estimated by:

$$Gr = \frac{M_{\text{SiC}}}{\rho_{\text{SiC}}}\left(j_{dep} - j_{ev}\right) \tag{4}$$

using the atomic carbon effective flux only. In Equation (4), $\rho_{\text{SiC}}$ is the SiC density and $M_{\text{SiC}}$ the SiC molar mass, while the expressions for $j_{dep}$ and $j_{ev}$ are given by

$$j_{dep} = j_{dep}(\text{C}) = j_{dep}(\text{Si}_2\text{C}) + 2 \times j_{dep}(\text{SiC}_2) =$$

$$= \sqrt{\frac{1}{2\pi M_{\text{Si}_2\text{C}}RT_{Source}}}exp\left[\frac{A_{\text{Si}_2\text{C}}}{T_{Source}} + B_{\text{Si}_2\text{C}}\right] + 2\sqrt{\frac{1}{2\pi M_{\text{SiC}_2}RT_{Source}}}exp\left[\frac{A_{\text{SiC}_2}}{T_{Source}} + B_{\text{SiC}_2}\right] \tag{5}$$

$$j_{ev} = j_{ev}(\text{C}) = j_{ev}(Si_2C) + 2 \times j_{ev}(SiC_2) =$$

$$= \sqrt{\frac{1}{2\pi M_{\text{Si}_2\text{C}}RT_{Sub}}}exp\left[\frac{A_{\text{Si}_2\text{C}}}{T_{Sub}} + B_{\text{Si}_2\text{C}}\right] + 2\sqrt{\frac{1}{2\pi M_{\text{SiC}_2}RT_{Sub}}}exp\left[\frac{A_{\text{SiC}_2}}{T_{Sub}} + B_{\text{SiC}_2}\right] \tag{6}$$

where $T_{Sub}$ and $T_{Source}$ are the substrate and the source temperatures, $M_{Si_2C}$ and $M_{SiC_2}$ are the molar masses of the Si$_2$C and SiC$_2$ molecules in the vapor phase, while $A_X$ and $B_X$ (whose values are such that the exponential terms have units of pressure [34]) are the experimental parameters which rule the partial pressures of the $X$ species in the vapor mixtures at thermodynamic equilibrium with the solid counterpart. $R$ is the universal gas constant. Different calibrations for the partial pressure-related parameters can be found in the literature, and the calibration impact will be described below.

Figure 6 shows the predicted growth rates obtained from Equation (4). The two-dimensional maps highlight the growth rate variation (color scale) at the middle position of the substrate (also indicated as seed) as a function of the source temperature (x-axes) and the temperature difference between source and seed at the middle-substrate position (y-axes). Two different calibrations for experimental vapor pressures of cubic SiC have been tested, one from Avrov's work [34] and one from Lilov [48]. Both vapor pressure datasets agree with each other. The simplified growth model of Equation (4) correctly predicts the experimental growth rates for the five different processes reported in Table 1.

**Table 1.** Experimental and simulated growth rates. The samples listed here refer to the growth conditions marked with A,B,C,D,E in Figure 6.

| Growth Rates [μm/Hours] | Sample A | Sample B | Sample C | Sample D | Sample E |
|---|---|---|---|---|---|
| Experimental | 101 | 208 | 230 | 231 | 287 |
| KMC MulSKIPS | 119 | 164 | 170 | 183 | 196 |

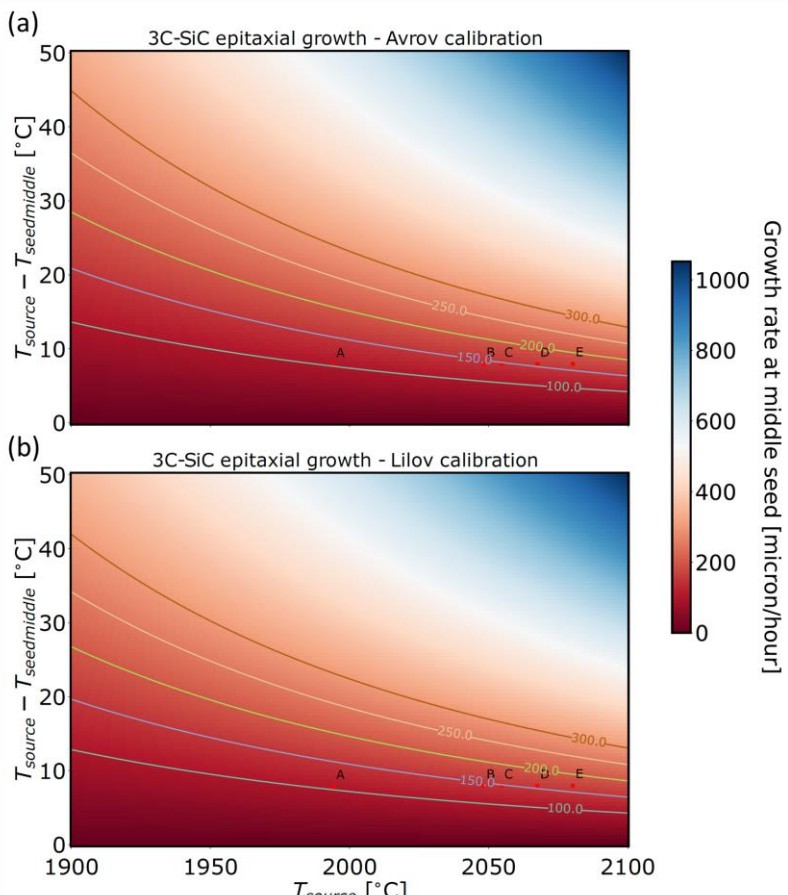

**Figure 6.** Two-dimensional map showing the growth rate variation (color scale) at the middle substrate position as a function of the source temperature (x-axes) and the temperature difference between the source and the seed at the middle-seed position (y-axes). Left (**a**) (Avrov [34]) and right (**b**) (Lilov [48]) panels refer to two different calibrations for the vapor gas pressures in the proximity of a cubic SiC substrate. The letters A,B,C,D,E indicate the growth conditions considered in this work for experiments and simulations (see Table 1). Isolines are also drawn to guide the eye.

### 3.2.2. Material Transformations at the Mesoscale in a PLA Process

PLA is applied to induce ultra-fast heating/cooling cycles in the processed samples, which is localized in the surface or in the first buried absorbing layer of a composite system. The laser fluence is chosen in a proper process window interval in order to obtain peak temperatures which usually exceed largely, in ten-hundreds ns wide intervals, the threshold for activating the desired internal material kinetics in the nanometric region heated by a laser. A typical process setting is the melting PLA where the locally confined melting is the key material modification; however, the melting is not mandatory for the use of PLA, since in non-melting conditions at high temperatures, a fast and relevant microstructural evolution can also occur. Examples include an important reorganization of defective regions, intermixing of solid alloys, and even phase transitions [49–51]. As already noticed, mesoscale modeling of these complex kinetics can be obtained with "equivalent" continuum and particle-like formalisms (see Ref. [36] for some examples of the two approaches). Here, we focus on models of the phase-field class which are particularly suited to simulate PLA at the mesoscale.

The phase field models allow an extension of the self-consistent solutions for the electro-magnetic and temperature fields, discussed in the previous section, to phase-field variables simulating internal regions of the material where phase transformations occur.

An example is the phase field equation simulating the evolution of the molten regions in the case of a melting process coupled with a modified expression for the heat equation:

$$\tau_\varphi \times \frac{\partial \varphi}{\partial t} = W^2 \nabla^2(\varphi) - 2\varphi(1-\varphi)(1-2\varphi) - 8\,\lambda(T,\varphi)[T-T_m]\varphi^2(1-\varphi)^2 \qquad (7)$$

$$c\frac{\partial T}{\partial t} = \nabla[k(T,\varphi)\nabla T] + 30\varphi^2\left(1-\varphi^2\right)L\frac{\partial \varphi}{\partial t} + S \qquad (8)$$

where $\varphi$ is the phase, $\tau_\varphi$ is the characteristic time of attachment of atoms at the interface, $W$ is the interface thickness, $S$ is the self-consistently computed heat source and $\lambda(T,\varphi)$ is a phase field function which implements the speed law as a function of the under/over cooling [52]. $\lambda(T,\varphi)$ is calibrated with the Karma–Rappel prescription: i.e., imposing that the diffuse interface model correctly reproduces the sharp interface limit and the latent heat $L$ balance at the moving interface [53]. The non-linear phase-dependent terms in Equations (7) and (8) are related to the particular phase field formulation applied (see ref. [42] and references therein for the derivation) with a phase function which recovers the liquid (solid) properties at the $\varphi = 0$ ($\varphi = 1$) value. All the material parameters in general could depend on temperature, atomic fraction for many atoms systems and phase [54]. The time $\tau_\varphi \sim$ ns and length $W \sim$ nm phase field parameters rule the physics of the phase transition region as the size- and the shape-related capillarity effects [53]. We note that, while environmental and material variables both have rule by master equations in the PDE class, the space resolution of the solution is completely different, since the former vary on a significantly larger scale with respect to the latter. However, the FEM methodology allows for suitable meshes' generation with mixed resolution: locally high resolution in the regions where the phase-field model acts and coarse in the regions where the environmental variables vary.

The phase field formalism permits large versatility in integrating case-dependent features, allowing the implementation of complex physical–chemical phenomena. A case of interest for the application in micro- and nano- electronics is the transition metal (e.g., Ni, Pt, Ti) silicide or germanide kinetics for the contact formation in devices based on group IV materials (Si, Ge, SiC). For such systems, due to fast inter-diffusion of combined atomic species, the formation of silicide compounds at the time scale of few tens of ns can be activated in a PLA process in both non-melting and melting modes [55,56]. A full model of the silicide formation kinetics can be formalized by expanding the phase-field equations in the alloy case (i.e., also considering an alloy-fraction-dependent melting point) and by coupling this extended model with a diffusion–reaction model for the prediction of the formation kinetics of silicide compounds (see, e.g., Refs. [50,56]).

The simulated evolutions evidence a structured scenario which has been validated by the post-process characterization of the corresponding processes. For increasing value of the fluence, different stages occur (sub-melting, partial melting and full melting of the metal rich film) with the proper characteristics, whereas binary compounds stabilize with the proper weights and space distribution [57,58]. An example of the model solutions for these processes (see Section 2) is reported in Figure 7, where the density of the silicide ($Ni_xSi_y$) compounds forming at the Ni-SiC interface is shown as a function of the time for a PLA process with a pulse of about 160 ns ($\lambda = 308$ nm) and 3.6 J/cm$^2$ energy density. The snapshots report a simulation analysis during the melting and regrowth stages of a full melting process, and the phase function is also plotted in these figures. In this case, the $Ni_3Si$, $N_5Si_2$ classes and $Ni_2Si$ compounds have similar weights in the silicide layer, which extends for 49 nm from the SiC interface that is displaced with time with respect to the original location due to the compound formations. We notice the formation of the $Ni_3Si$ compound in the solid region before the melt front reaches the interface depth, whilst the stability of the other compounds is obtained during the regrowth, also triggered by Ni-Si-C fast intermixing in the liquid phase.

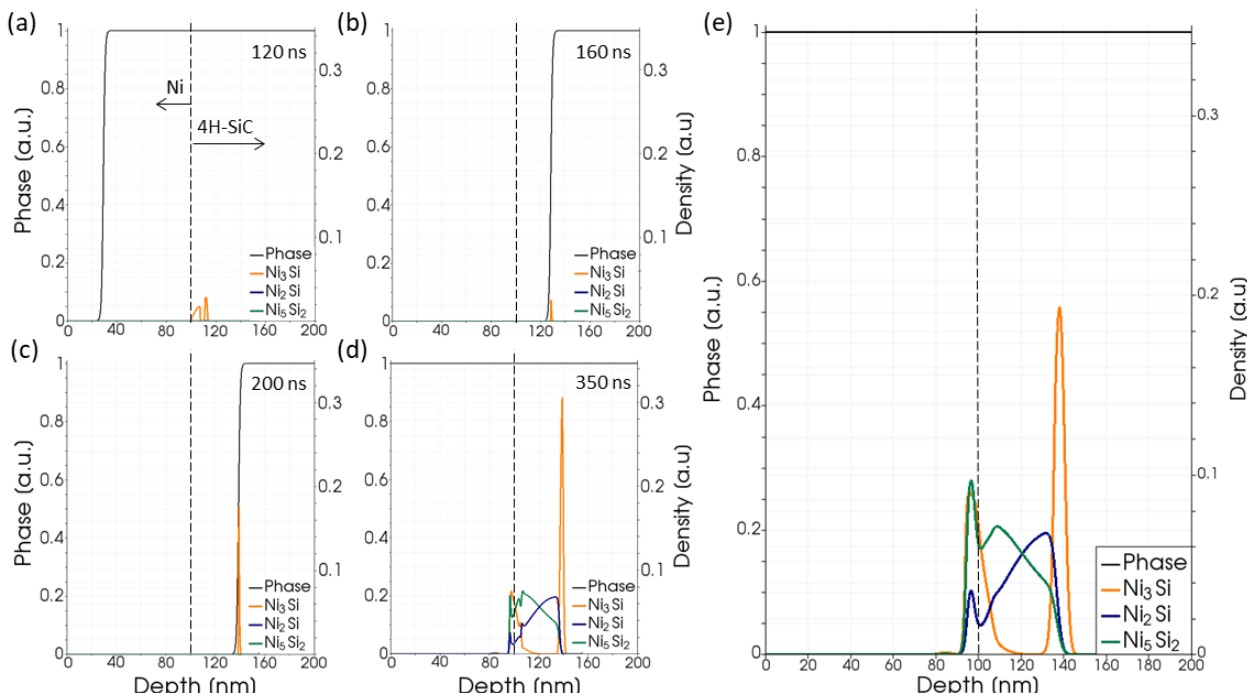

**Figure 7.** Phase field (black line and left axis with interface parameters $\tau_\varphi = 4.83 \times 10^{-9}$ s and $W = 1$ nm) and simulated local density (right axis scale) of the $Ni_3Si$ (dark yellow lines), $Ni_5Si_2$ class (dark green lines) and $Ni_2Si$ (blue lines) for a PLA laser annealing process with $\Delta t_{pulse} = 160$ ns and fluence of 3.6 J/cm$^2$. Snapshots (**a**–**d**) are taken at t = 120, 160, 200, 350 ns, while (**e**) is taken at the end of the simulation. The phase function $\varphi$ is plotted as a black line. The initial system is a deposited 100 nm thick Ni film on a 4H-SiC substrate (the original interface depth is indicated by a dashed black line) [58].

From this particular example, merits but also limitations of the mesoscale model are evident: after a careful calibration, average quantities can be simulated with some accuracy, but no information can be obtained in terms of, e.g., the impact of the crystalline structure of the silicide compounds which coexist in the same volume. In the next section, we will discuss how atomic scale details, required to account for these effects, can be properly included in the digital twins.

### 3.3. Process Simulation Predictions at the Atomic Scale

Mesoscale modeling can allow for an accurate estimation of average process quantities and can span space/time scales of real experiments. However, thermally activated processes often induce local morphological alterations in the manipulated materials that are important for their functionality when implemented in devices and applications. For example, both PVD growths and PLA treatments can generate point and extended defects in the processed materials [59,60], that could be detrimental under device operation, e.g., giving rise to leakage currents or reducing the mobility of electrons and holes. In this case, atomic-scale modeling becomes fundamental for the correct description of the local changes induced by these processes on the manipulated systems. Atomistic simulation techniques can vary in the description of atomic interactions and can be usually split into ab initio and parameterized methods, with the former being more accurate in the description of chemical bonding and the latter being more efficient in the simulation of large systems for long time scales. The multiscale approach allows for a combination of both approaches by obtaining parameters from first principle calculations and applying them in optimized process simulation techniques, such as the Lattice Kinetic Monte Carlo (KMC) method. Within this context, this section will discuss the atomistic implementation of process simu-

lation for both PVD and PLA, based on multiscale implementations of specifically adapted KMC approaches.

### 3.3.1. Defect Generation and Evolution in a 3C-SiC PVD Growth

As an example of atomistic PVD growth, we studied the generation of extended defects in 3C-SiC using a super-Lattice KMC methodology that is able to follow the kinetic evolution of $sp^3$-type crystals, including the presence of defects that break the stacking sequence of the cubic polymorph, introducing hexagonal inclusions within the crystal. This aspect is fundamental for the correct evaluation of extended defects such as stacking faults and antiphase boundaries, which are commonly present in the material. The proposed KMC methodology is based on deposition and evaporation Monte Carlo events that are driving the stochastic evolution of the system by means of independent kinetics for silicon and carbon atoms. These events are computationally described by Arrhenius-type functions with activation energies and frequency pre-factors that can be calibrated from ab initio calculations and from the equilibrium partial pressures in the gas phase, respectively. Computations presented below are performed with the open-source MulSKIPS code [17,18], which allows the initialization of the geometry though standard TCAD tools.

The atomic fluxes of the PVD process are set according to the discussion of Section 3.1. The typical workflow for the KMC simulation of the epitaxial growth of a (100) cubic silicon carbide surface contains the following steps:

1.  First, the growth rate of the process is estimated from the experimental gas vapor pressures.
2.  Consequently, the input file for the simulation of the PVD epitaxial growth within the MulSKIPS code is generated. This contains information on the simulated structure, the energetics of the Monte Carlo events, as well as their frequency pre-factors.
3.  Then, the MulSKIPS simulation runs in a pre-defined folder.
4.  At the end of the simulation, analysis of the crystalline quality of the epitaxial-growth substrate and the eventual epitaxial defects takes place.
5.  Finally, the growth rate is extracted from the KMC MulSKIPS run and can be readily compared to the respective experimental results.

We have performed simulations equivalent to five experimental epitaxial processes (see Figure 5) that are characterized by defect-free flat substrates. Figure 8 shows a KMC MulSKIPS simulation with process conditions fixed to those of the experimental processed sample A. Figure 8a shows the time evolution of the surface average height, whilst Figure 8b shows the growth rate variation. Both figures indicate a constant epitaxial growth of the (100) cubic SiC surface. The experimental and KMC growth rates are reported in Table 1. Their comparison suggests that in both cases, trends in the growth rates with varying process conditions agree consistently.

As discussed earlier, the MulSKIPS code separately considers the Si- and C-based atomic elementary evolution event (namely deposition or evaporation). As a consequence, both C and Si species fluxes, present as abundant species in the SiC vapor in contact with the solid phase, are included using the expressions of Equations (5) and (6).

A study of the evolution from the initial system, where a defective nucleus is present before the successive PVD growth, has been performed after an extension of the initialization routines of the MulSKIPS simulations, including the possibility of setting up a triple Staking Fault (SF) defect (micro-twin) in the (100) substrate. In Figure 9a, the initial system is shown: the defect habit plane is {111}, while the borders of the triangular-shaped defect are oriented in directions <110>. The length of the intersection of the defect's nucleus and the initial substrate surface is 22 nm. In this study, six replicas (i.e., simulations that are statistically equivalent using different seeds for the random number generator) of the simulated evolution at each of the five growth conditions already presented are used to characterize the SF. In Figure 9b,c, two snapshots of the simulated evolution after the simulation of a $\approx 50$ nm thick grown layer are shown. In all the 30 simulated cases, as in the two reported in the figure, we observe that the initial planar defect tends to expand

during the growth. These results should indicate an intrinsic kinetic stability of micro-twin defects in the case of missing interference due to the interaction with other defects or with some particular evolving surface morphology (e.g., steps or shape change). We notice that the surface remains flat during all the simulated growths.

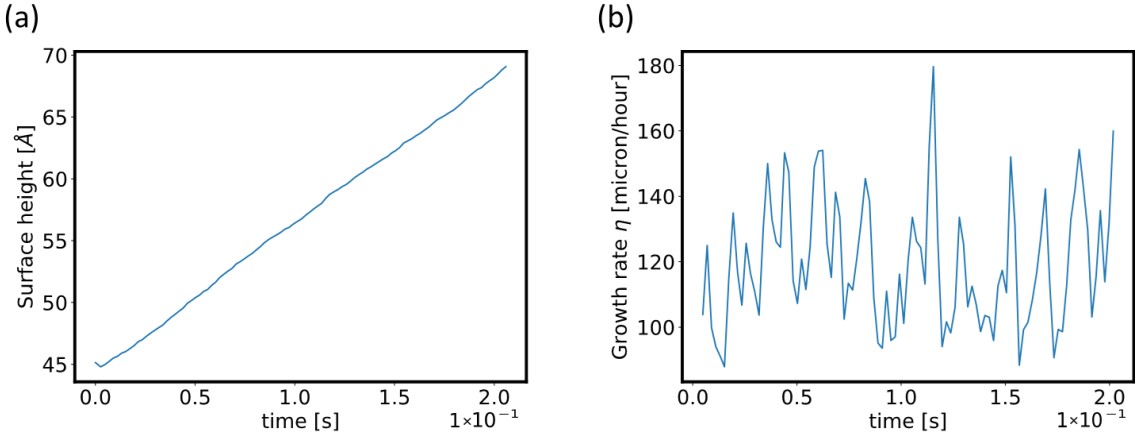

**Figure 8.** KMC MulSKIPS simulation with process conditions fixed to those of the experimental processed Sample A: (**a**) Time evolution of the surface average height; (**b**) growth rate variation.

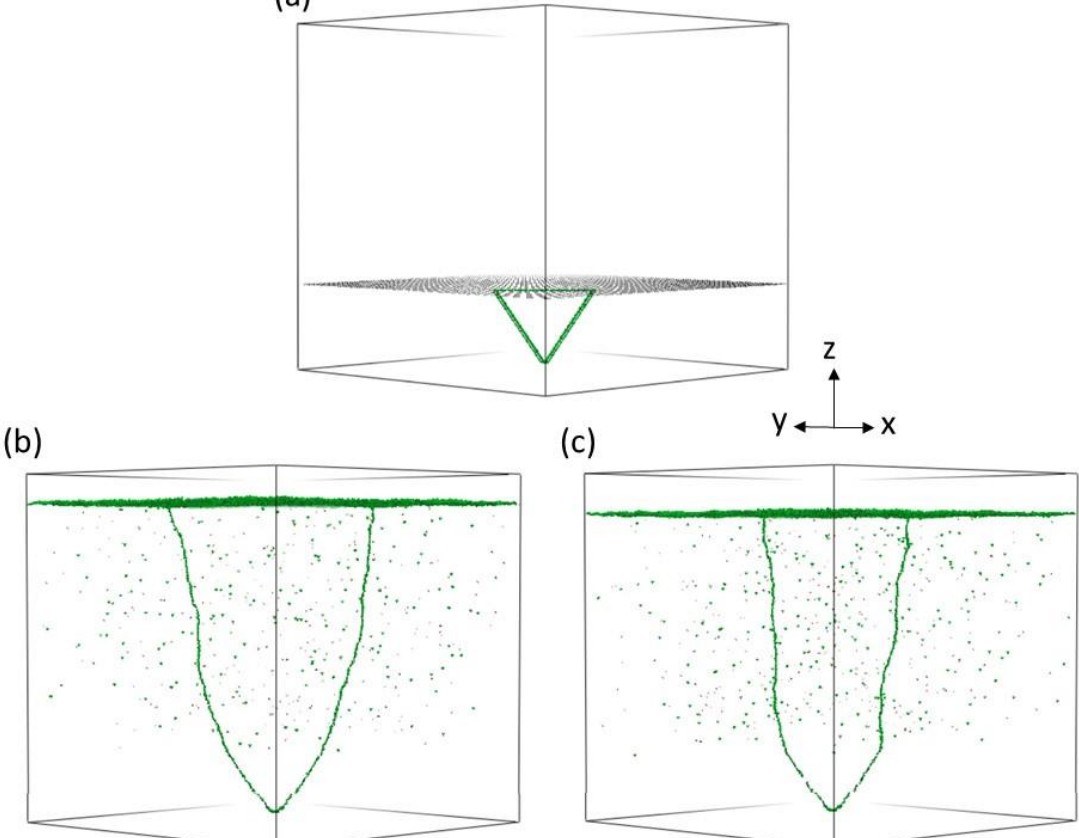

**Figure 9.** Simulation of SiC PVD growth starting from a defective nucleus. (**a**) Initial SF nucleus. Only uncoordinated atoms are shown (i.e., vacancies, SF borders and surface). Si atoms green spheres, C atoms gray spheres. Snapshot of the KMC-MulSKIPS simulated evolution of triple SF defects for the growth conditions of (**b**) sample A and (**c**) sample E (i.e., the lowest and the highest growth rate among the 5 conditions discussed here) obtained after similar thicknesses of the grown layer. Only uncoordinated atoms are shown (i.e., vacancies, SF border and surface).

As a result of the expansion, the length of the intersection of the defect's nucleus and the initial substrate surface ($L_{SF}$) increases. In Figure 10, we plot the average value of the $L_{SF}$ obtained in the six simulated replicas after a $\approx 50$ nm thickness of growth for the SiC layer. The eventual differences between lower and higher values of the source temperature are relatively small, and they should be confirmed by additional extended statistical analyses.

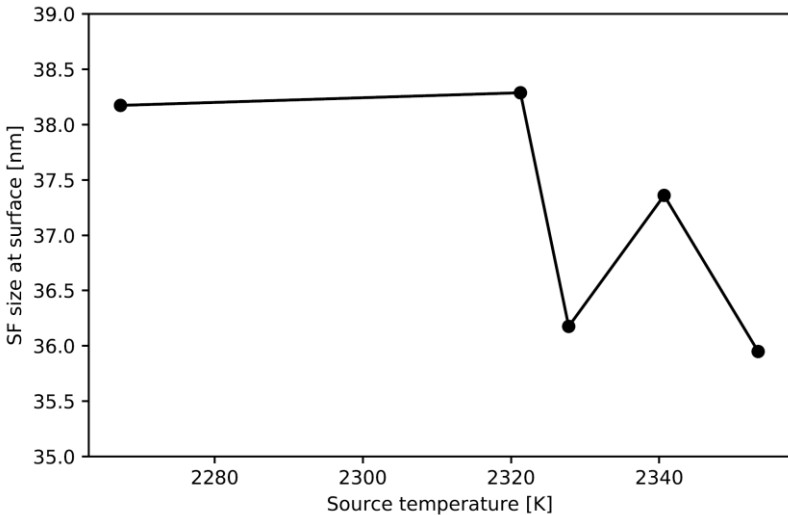

**Figure 10.** Average intersection length of the SF at the (100) surface as a function of the PVD growth conditions. For all the cases, growths of $\approx 50$ nm-thick layers are considered.

The example above emphasizes the importance of atomistic simulations for growth processes when the presence of local or extended defectiveness is essential for the description of the structural characteristics of the grown material and could have significant implications in its use in devices and applications.

### 3.3.2. FEM–KMC Modeling of PLA of a Si(001) Surface

The continuum description of materials, typical of most state-of-the-art PLA process simulators, has a fundamental limitation when it comes to predicting the evolution of the system with nano-scale resolution, which is crucial to locally tailor its structure and properties towards a specific sought functionality. For instance, the morphology of liquid droplets or the quenched surface generated during the laser-induced ultra-rapid melting phenomenon are deeply bound to the lattice symmetries of the annealed surface and the highly crystal-orientation-dependent kinetics of individual atoms [61–63]. A reliable PLA virtual twin should be able to capture all relevant atomic-scale phase changes while correctly modeling radiation interaction with matter and the variations in the thermal field self-consistently over the whole pulse duration.

In this context, an efficient multiscale approach consists of coupling a FEM mesoscale solver for Maxwell and thermal diffusion equations with a super-lattice KMC simulator of melting/quenching within a local nanoscale portion of the system [21]. The algorithm is schematically illustrated in Figure 11 in the case of PLA of a Si(001) surface. It starts with setting up the FEM continuum mesh with a specific system size and composition. The electromagnetic–thermal problem is solved until local surface melting begins, triggering a self-consistent procedure where the FEM solver works in parallel with the KMC simulator of the melting region. For the whole laser pulse duration, and with sub-nanosecond time resolution, (1) the thermal field is interpolated from the FEM mesh to the KMC super-lattice; (2) the KMC simulates the nucleation kinetics and the liquid front evolution, reproducing potential atomic-scale structural modifications, defects, segregation, stress generation and relaxation, etc.; (3) the solid/liquid volumes in the FEM are updated and impact the thermal field evaluation in the subsequent iteration.

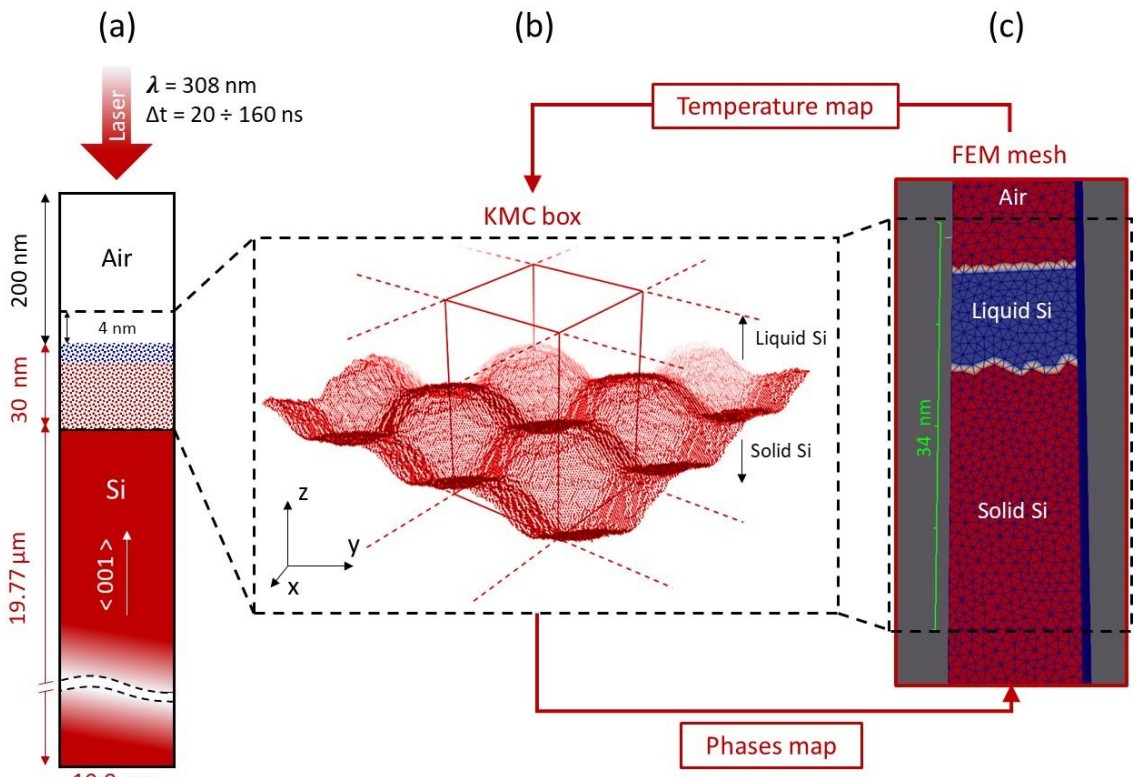

**Figure 11.** Schematics and validation of the multiscale FEM-KMC method for PLA simulations. (**a**) Outline of input FEM geometry for Si(001), with dotted area indicating the sub-system modeled with KMC (blue: liquid Si volume; red: solid Si volume). (**b**) KMC box, containing the solid–liquid front evolving during irradiation (green: under-coordinated solid Si atoms). (**c**) Sub-region in FEM mesh coupled with KMC (blue: liquid Si volume; red: solid/air volumes).

The accuracy of such approach can be inferred from Figure 12a, which shows the maximum melt depth as a function of the laser fluence calculated by assuming a PLA process of the Si(001) surface with 308 nm laser wavelength and 160 ns pulse duration. The obtained results are in excellent agreement with both simulated data from a fully continuum 1D phase-field model, and measured data from secondary-ion mass spectroscopy experiments.

As an example, the considered multiscale framework can be used to predict how an in-homogeneous molten phase at the earliest stage of melting would evolve at the atomic level during the ultrafast irradiation (see Figure 12b–e). It is found that an initial array of hemispherical liquid Si nuclei (Figure 12b) can either quench or reshape into highly symmetric tetrahedral nanodroplets (Figure 12c), and even coalesce into a rough liquid layer (Figure 12d,e) before full re-solidification, all depending on the particular value of laser energy density set up in the macroscale FEM environment.

It can be noted that, in principle, this formalism can be extended to any situation where the system kinetics at the sub-nanometer scale is determined by temperature, strain or any space (time) dependent fields existing in the meso (macro) scale environment. An example of code implementing such approach is distributed open source with MulSKIPS [17,18], which is readily applicable to group IV semiconductors, provided a proper calibration and is generalizable to other crystals, including compounds or metals, upon redefinition of the KMC lattice symmetries. Interestingly, formation and evolution of defects can also be accounted for in the simulation, which is of great interest for reliable morphology and quality predictions of a potential PLA digital twin.

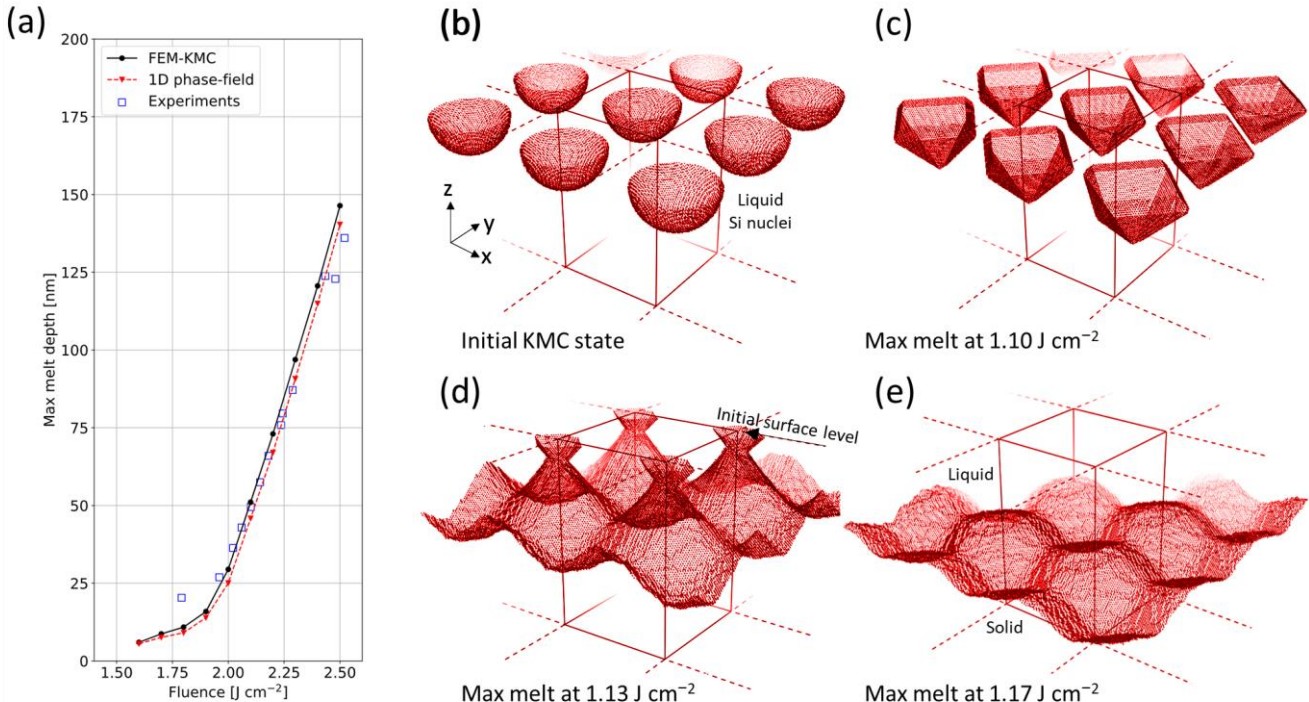

**Figure 12.** FEM-KMC simulations of Si(100) PLA. (**a**) FEM-KMC calculated maximum melt depth as a function of laser fluence for a PLA of Si(001) with a 308 nm and 160 ns laser pulse, in comparison with equivalent results from 1D phase-field simulations as well as experimental data. (**b**) Periodic visualization of a KMC box showing the solid–liquid interface (green), embedding the (initially hemispherical) liquid Si nano-droplet. (**c**) Reshaped nuclei at 1.10 J/cm$^2$ fluence at the time $t_{MMV}$ of maximum melt volume in the simulation (after 41 ns of irradiation). (**d**) Melting front at $t_{MMV}$ (37 ns) at 1.13 J/cm$^2$, with nuclei partially coalesced. (**e**) Melting front at $t_{MMV}$ (32 ns) at 1.17 J/cm$^2$, with nuclei completely coalesced into a rough molten layer.

## 4. Discussion

As discussed in the introduction, digital twins are "in-silico" manufacturing lines, which, for complex production systems such as the ones implemented for the fabrication of micro- and nano- electronic devices, cannot solely rely on chemical–physical simulation tools. Indeed, approaches belonging to the class of AI, whose predictivity is gained by ("machine") learning from the databases accumulated with the previous use of the processes in the line, are also fruitfully used for digital twin development [3]. Moreover, AI can be also implemented at a superior level for the holistic management of the whole manufacturing line [64].

If we restrict our considerations to a particular process module (etching, thermal processes, depositions, lithography, etc.), VM/AI and chemical–physical simulations can be used alternately or in combination to predict the single process results (digital twin module; see Figure 1). The pros and cons of the two methods are clear. VM is fast and generalizable, but it lacks any intrinsic predictivity potential and it can barely be applied from scratch far away from the previously explored settings. The development of simulation tools dedicated to complex processes represents a difficult challenge for computational materials science, but it can gain intrinsic predictivity by connecting the process parameters to atomic-scale modifications of the fabricated nano-objects.

In this paper, we discussed two digital twin modules based on multiscale simulation approaches which bridge macro-, meso- and atomic- scale evolutions, which concur to the determination of the final process results. We indicated the suitable formalisms for each scale, and how numerical methods for the solution of the models could be coupled to achieve a complete and effective virtualization of the process. Although the specific examples discussed in this work are limited to systems with sp$^3$ symmetry and the need for

proper calibrations based on ab initio inputs or experimental validations, we believe that the numerical strategies implemented in these specific examples can be properly generalized to other cases and materials. Moreover, to support this statement, we notice that although the physical phenomena underlying the two processing examples are completely different, the simulation methods are already integrated in the same open-source tool [17]. This is a promising achievement for the development of future simulations based on digital twin modules, which spurs further advancements not only in the context of materials' manufacturing processes, but also in that of advanced control at equipment level (e.g., reactors, etc.), carriers transport in nanodevices, for which multiscale simulation strategies are in constant evolution [65,66], as well as in the broader context of Industrial Internet of Things (IIoT) and Digital Factories [67]. Overall, although the development effort of multiscale methodologies is obviously significantly more intense with respect to the one necessary for black-box VM techniques, we believe that the benefit in terms of basic understanding and accurate process control in any application condition could outweigh this cost.

**Author Contributions:** Conceptualization, A.L.M.; methodology, G.C. and A.L.M.; investigation and analysis, G.C., G.F., A.L.M., M.K., M.S., P.J.W., S.F.L. and F.L.V.; writing, G.C., G.F., I.D., A.L.M., M.K., M.S. and P.J.W.; funding acquisition, A.L.M. All authors have contributed to reviewing and editing the manuscript. All authors have read and agreed to the published version of the manuscript.

**Funding:** G.C., I.D, G.F., A.L.M. gratefully acknowledge funding from the European Union's Horizon 2020 Research and Innovation programme under grant agreement No. 871813 MUNDFAB.

**Data Availability Statement:** The data that support the findings of this study are available from the corresponding author, A.L.M., upon reasonable request.

**Conflicts of Interest:** The authors declare no conflict of interest. The funders had no role in the design of the study; in the collection, analyses, or interpretation of data; in the writing of the manuscript; or in the decision to publish the results.

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
