# Peer review of "Multiscale Simulations for Defect-Controlled Processing of Group IV Materials"

_crystals, doi:10.3390/cryst12121701_

Round 1

Reviewer 1 Report

In this manuscript, the author discusses two digital twin modules based on multi-scale simulation methods, which link the macro-, meso- and atomic- scale evolution to determine the final process results. For these two special research cases, the possibility of simulating defect generation and evolution in atomic resolution during the whole process duration is provided. It is the reviewer's opinion that the solution methodology is sound and the topic is hot and novel. In this sense, the reviewer suggests this manuscript may be published in this journal after addressing some revisions.

(1) The abstract should be organized and the research content of the article should be highlighted.

(2) The color and configuration of Figure1 should be optimized.

(3) Could 'critical data' or 'sub critical data' in Figure 1 be explained more specifically?

(4) The author should explain more details about the multi-scale prediction process.

(5) Is it more appropriate to use the inclusion as the title in section 4?

(6) The limitations of this study should be exposed.

(7) This article needs to improve the quality of the illustrations, especially the size and clarity of the fonts in the Figures.

Reviewer 2 Report

The usage of AI for the creation of the digital twin could be elaborated more, if authors have relevant information.

Reviewer 3 Report

The manuscript presents a masterful view of multiscale simulations of group IV materials (and possibly extendible to other materials) centred around meso-macro systems.  While I am not an expert in "COMSOL" finite-element and such engineering techniques (my world is that of electronic structure, trying to stretch to the nano-micro regime of nanocatalysis, on the one hand, and enzymes on the other) I found the manuscript to present a compelling picture of how to bring techniques together from the atomistic to the macro scales (through kinetic monte carlo). I thoroughly enjoyed studying this manuscript.  

I think it could be published as is, but I might suggest that the authors' "stretch" is actually a little larger than atomistic, at the lower end. In fact, in section 3.3.1 the authors use ab initio calculations to calibrate activation energies and frequencies.  The latter are in the "electronic" scale, which I would say, is distinguishable from "atomistic" (such as force-field molecular dynamics).  But I leave it up to the authors whether they would like to make this extra claim

At the other end, I wonder if the authors could speculate a little on how to extend their studies to even larger scales.  For example, an economics analysis comes to mind? perhaps putting the nano-devices in the context of the factory.  But, again, it is entirely up to the authors how much they would like to add a "future perspective" component to their paper.

Reviewer 4 Report

file attached
